# Availability, affordability and access to essential medications for asthma and chronic obstructive pulmonary disease in three low- and middle-income country settings

Trishul Siddharthan[1,2,3]*, Nicole M. Robertson[3,4], Natalie A. Rykiel[2,3], Lindsay J. Underhill[2,3], Nihaal Rahman[3], Sujan Kafle[5], Sakshi Mohan[6], Roma Padalkar[7], Sarah McKeown[8], Oscar Flores-Flores[8,9], Shumonta A. Quaderi[10], Patricia Alupo[11], Robert Kalyesubula[12], Bruce Kirenga[11], Jing Luo[13], Maria Kathia Cárdenas[14], Gonzalo Gianella[14], J. Jaime Miranda[14,15], William Checkley[2,3,8], John R. Hurst[10], Suzanne L. Pollard[2,3,8]

1 Division of Pulmonary and Critical Care, School of Medicine, University of Miami, Miami, Florida, United States of America, 2 Division of Pulmonary and Critical Care, School of Medicine, Johns Hopkins University, Baltimore, Maryland, United States of America, 3 Center for Global Non-Communicable Disease Research and Training, Johns Hopkins University, Baltimore, Maryland, United States of America, 4 University of Kentucky School of Medicine, Louisville, Kentucky, United States of America, 5 Institute of Medicine, Tribhuvan University, Kathmandu, Nepal, 6 Centre for Health Economics, University of York, York, United Kingdom, 7 Rowan University School of Osteopathic Medicine, Glassboro, New Jersey, United States of America, 8 Department of International Health, Bloomberg School of Public Health, Johns Hopkins University, Baltimore, Maryland, United States of America, 9 Universidad de San Martin de Porres, Facultad de Medicina Humana, Centro de Investigación del Envejecimiento (CIEN), Lima, Peru, 10 Respiratory Medicine, University College London, London, United Kingdom, 11 Makerere Lung Institute, Makerere University, Kampala, Uganda, 12 College of Health Sciences, Makerere University, Kampala, Uganda, 13 University of Pittsburgh, Pittsburgh, Pennsylvania, United States of America, 14 CRONICAS Centre of Excellence in Chronic Diseases, Universidad Peruana Cayetano Heredia, Lima, Peru, 15 School of Medicine, Universidad Peruana Cayetano Heredia, Lima, Peru

* tsiddhar@miami.edu

**Data Availability Statement:** Datasets have been uploaded as supplementary information.

## Abstract

### Introduction

Despite the rising burden of chronic respiratory disease globally, and although many respiratory medications are included in the World Health Organization Essential Medications List (WHO-EML), there is limited information concerning the availability and affordability of treatment drugs for respiratory conditions in low- and middle-income countries (LMICs).

### Methods

All public and private pharmacies in catchment areas of the Global Excellence in COPD outcomes (GECo) study sites in Bhaktapur, Nepal, Lima, Peru, and Nakaseke, Uganda, were approached in 2017–2019 to assess pricing and availability of medications for the management of asthma and COPD.

### Results

We surveyed all 63 pharmacies in respective study areas in Nepal (95.2% private), 104 pharmacies in Peru (94.2% private) and 53 pharmacies in Uganda (98.1% private). The

**Funding:** The authors received no specific funding for this work.

**Competing interests:** The authors have declared that no competing interests exist.

availability of any medication for respiratory disease was higher in private (93.3%) compared to public (73.3%) pharmacies. Salbutamol (WHO-EML) monotherapy in any formulation was the most commonly available respiratory medication among the three sites (93.7% Nepal, 86.5% Peru and 79.2% Uganda) while beclomethasone (WHO-EML) was only available in Peru (33.7%) and Nepal (22%). LABA-LAMA combination therapy was only available in Nepal (14.3% of pharmacies surveyed). The monthly treatment cost of respiratory medications was lowest in Nepal according to several cost metrics: the overall monthly cost, the median price ratio comparing medication costs to international reference prices at time of survey in dollars, and in terms of days' wages of the lowest-paid government worker. For the treatment of intermittent asthma, defined as 100 mcg Salbutamol/Albuterol inhaler, days' wages ranged from 0.47 days in Nepal and Peru to 3.33 days in Uganda.

## Conclusion

The availability and pricing of respiratory medications varied across LMIC settings, with medications for acute care of respiratory diseases being more widely available than those for long-term management.

## Introduction

The burden of chronic respiratory disease is increasing globally, with the highest disease-related morbidity and mortality ocurring in low- and- middle-income countries (LMICs). Obstructive respiratory disease such as asthma and chronic obstructive pumonary disease (COPD) affect an estimated 174 million and 358 million individuals worldwide, respectively [1, 2]. While asthma and COPD age-standardized death rates have decreased over the past two decades, the majority of these gains have occurred in high-income countries [1]. Individuals in LMIC settings have unique risk factors for chronic respiratory conditions, including household and ambient air pollution, biomass exposure, rapid urbanization, occupational exposure, and prior communicable disease such as pneumonia and TB [3]. Furthermore, significant barriers exist in these settings for effective management of chronic respiratory diseases [4]. These include limited public health literacy pertinent to these conditions, insuffient diagnostic equipment and respiratory specialists, and importantly, inadequate availability and affordability of medications necessary for chronic disease management [5, 6]. As the prevalence of chronic respiratory diseases increase, the lack of access to effective management and essential medications for chronic respiratory diseases have resulted in significant increases in age-standardized disability-adjusted life years (DALYs) attributable to these conditions [1].

A number of studies have examined availability and affordability of essential medications for the management of COPD and asthma in LMIC settings, although these studies have been traditionally limited to a smaller selection of pharmacies in urban and rural centers [7]. Essential medications are defined as those that meet priority health care needs, have strong evidence of efficacy and safety, and are cost effective [6]. According to the World Health Organization (WHO), access to essential medications for noncommunicable diseases has been deemed a health care priority, with a goal of 80% availability of affordable technologies and essential medications to treat non-communicable diseases [8]. The WHO Essential Medicines List (EML) includes beclomethasone, budesonide, budesonide/formoterol, ipratropiom bromide and salbutamol (also known as albuterol) aerosolized therapy, and epinephrine injection, as

medications that are considered minimum medicinal needs for a basic health system [8]. Despite global consensus on the need for essential medications for chronic respiratory disease in LMICs, availability of essential medications is limited. When these medications are present at a pharmacy, they are often unaffordable [6, 9].

We surveyed all public and private pharmacies and health centers in three LMIC communities and assessed the availability and affordability of a comprehensive list of medications for the management of asthma and COPD [10]. While prior studies have been conducted in the region, none have assessed all pharmacies within catchment areas to ascertain medication availability beyond the EML, nor have they calculated the cost of therapy based on treatment guidelines utilizing all pharmacy information available [11]. We aimed to develop a comprehensive list of chronic respiratory disease medications to assess treatment costs for asthma and COPD.

## Methods

### Study design and setting

Global Excellence in COPD Outcomes (GECo) is a population-based study in three distinct geographic and economic regions in Asia, South America, and sub-Saharan Africa [10, 12]. We carried out a cross-sectional survey of public and private pharmacies (including health centers and informal pharmacies) in defined catchment areas in Bhaktapur, Nepal (2019), Lima, Peru (2017), and Nakaseke, Uganda (2019) (see Fig 1 and S1 Data) as part of the parent GECo Study to assess pricing and availability of medications for the management of asthma and COPD [10].

This study was reviewed and approved by the University College London Research Ethics Committee (9661/001), Johns Hopkins School of Medicine (IRB00139901), Uganda National Council for Science and Technology, Makerere School of Medicine (SOMREC 2017–096), Nepal Health Research Council (136/2017) and A.B. PRISMA (CE2147.17). Data on medication costs and availability were publicly available and no personal data were collected from pharmacy technicians or pharmacists who were contacted; for those reasons, informed consent was not obtained.

### Data collection

Trained data collectors visited all health centers and pharmacies in study areas and surveyed medicine availability and price using a modified version of the WHO-Health Action International Survey [13]. Surveys were conducted during an 8–16-week period between 2017 and 2019. Prices were adjusted based on year of data collection. Pharmacies were surveyed twice if the pharmacy was closed at first visit, or if the pharmacist was unavailable. Data on pharmacy type (public vs private), as well as GPS location and hours of operation were collected. Public pharmacies were defined as being government operated and having free or subsidized pricing for medication. We surveyed all medications used for the management of chronic respiratory disease including inhaled corticosteroids (ICS), short acting beta-agonist (SABA), long-acting beta-agonist (LABA), short-acting muscarinic agent (SAMA), long-acting muscarinic agent (LAMA), theophylline/aminophylline as well for treatment of exacerbations (steroids and antibiotics). Data on the management of asthma (certain SABA, ICS, LAMA, LABA) was collected in Peru, whereas additional data pertaining to COPD was collected in Uganda and Nepal (LAMA). Medications included in the survey for each country are listed in Table 1.

### Measurements

For each medication, we collected information on the drug presentation, the availability (available to purchase on the day of the visit) and affordability (retail price unit for each generic or

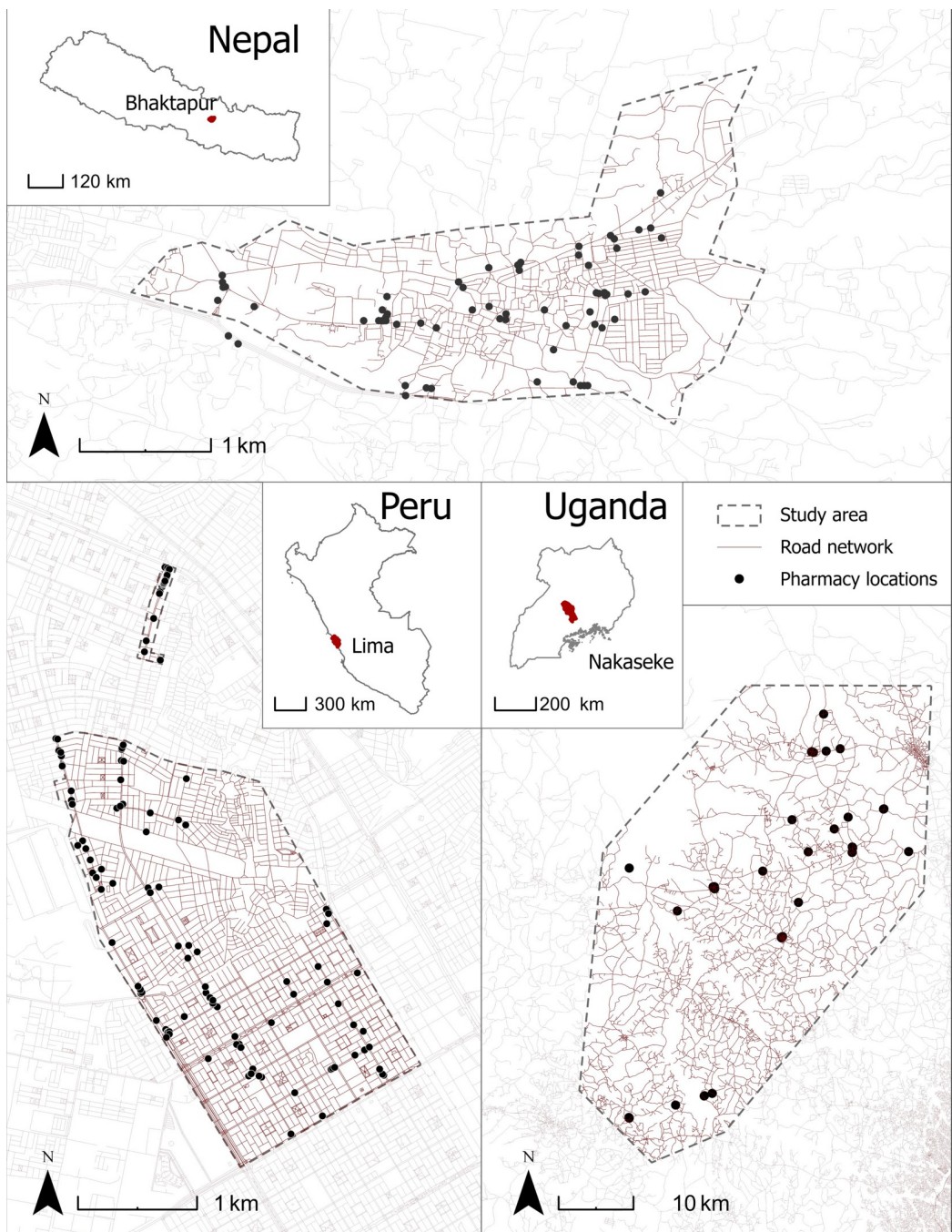

**Fig 1. Pharmacy locations in Bhaktapur, Nepal, Lima, Peru, and Nakaseke, Uganda.** All pharmacies surveyed within study catchment areas as indicated by GPS including road networks. OpenStreetMap contributors. (2019) Data Extracts. Retrieved from https://planet.openstreetmap.org. Global Administrative Areas. (2021) GADM database. Retrieved from https://gadm.org/data.html.

brand name formulation of the medications) [14]. Medications with different formulations (e.g. salbutamol, which comes in inhaler, pill, and syrup form) were asked about separately. Wages for the lowest unskilled government workers were queried from government records at the time of data collection (S1 Data).

**Table 1. Availability of respiratory medications in Bhakthapur Nepal, Lima Peru and Nakaseke Uganda by generic and brand name.** Grey indicates no available data.

| Medication Category | Medication | Frequency of Pharmacies Stocking n (%) | | | | | |
| --- | --- | --- | --- | --- | --- | --- | --- |
| | | Nepal (n = 63) | | Peru (n = 104) | | Uganda (n = 53) | |
| | | Generic | Brand | Generic | Brand | Generic | Brand |
| Antibiotics | Amoxicillin | 63 (100) | 0 | | | 38 (71.7) | 0 |
| | Amoxicillin +Clavulanic Acid | 54 (85.7) | 0 | | | 6 (11.3) | 2 (3.8) |
| | Ampicillin-Cloxacillin | 0 | 0 | | | 33 (62.3) | 0 |
| | Azithromycin | 60 (95.2) | 0 | | | 8 (15.1) | 0 |
| | Cefpodoxime | 30 (47.6) | 0 | | | 0 | 0 |
| | Cefpodoxime + Clavulanic Acid | 17 (27.0) | 0 | | | 0 | 0 |
| | Ciprofloxacin | 0 | 0 | | | 34 (64.2) | 0 |
| | Clarithromycin | 0 | 0 | | | 5 (9.4) | 0 |
| | Co-trimoxazole | 0 | 0 | | | 1 (1.9) | 0 |
| | Doxycycline | 36 (57.1) | 0 | | | 27 (50.9) | 0 |
| | Erythromycin | 0 | 0 | | | 22 (41.5) | 0 |
| | Levofloxacin | 26 (41.3) | 0 | | | 7 (13.2) | 0 |
| | Tetracycline | 0 | 0 | | | 13 (24.5) | 0 |
| Antihistamines | Cetirizine | | | 4 (3.8) | 2 (1.9) | | |
| | Chlorphenamine | | | 4 (3.8) | 2 (1.9) | | |
| | Loratadine | | | 2 (1.9) | 1 (1.0) | | |
| Inhaled Corticosteroids (ICS) | Beclomethasone | 14 (22.2) | 0 | 35 (33.7) | 1 (1.0) | 0 | 0 |
| | Budesonide | 26 (41.3) | 0 | | | 0 | 0 |
| | Fluticasone | 8 (12.7) | 0 | 0 | 6 (5.8) | 0 | 0 |
| ICS + LABA | Fluticasone + Salmeterol | 45 (71.4) | 0 | 3 (2.9) | 5 (4.8) | 0 | 0 |
| | Budesonide + Formoterolfuerate | 55 (87.3) | 0 | | | 0 | 0 |
| ICS + SABA | Beclomethasone + Levosalbutamol | 15 (23.8) | 0 | | | 0 | 0 |
| | Beclomethasone + Salbutamol | 0 | 0 | 2 (1.9) | 4 (3.8) | 0 | 0 |
| LABA | Salmeterol | 19 (30.2) | 0 | 6 (5.8) | 1 (1.0) | 0 | 0 |
| LAMA | Tiotropium | 49 (77.8) | 0 | | | 0 | 0 |
| Mucolytic | Acetylcysteine | 11 (17.5) | 0 | | | 0 | 0 |
| LABA + LAMA | Formoterol + Tiotropium | 9 (14.3) | 0 | | | 0 | 0 |
| Oral Corticosteroids (OCS) | Prednisolone | 40 (63.5) | 0 | 1 (1.0) | 11 (10.6) | 0 | 0 |
| | Dexamethasone | 25 (39.7) | 0 | 81 (77.9) | 55 (52.9) | 52 (98.1) | 0 |
| | Prednisone | 0 | 0 | 91 (87.5) | 49 (47.1) | 47 (88.7) | 0 |
| SABA | Salbutamol/Albuterol | 60 (95.2) | 0 | 90 (86.5) | 40 (38.5) | 42 (79.2) | 5 (9.4) |
| | Levosalbutamol | 3 (4.8) | 0 | 0 | 0 | 0 | 0 |
| | Salbutamol + Bromhexine | 58 (92.1) | 0 | | | 0 | 0 |
| | Terbutaline + Bromhexine | 59 (93.7) | 0 | | | 0 | 0 |
| | Salbutamol + Bromhexine + Etofylline | 10 (15.9) | 0 | | | 0 | 0 |
| SABA +xanthine | Salbutamol + Theophylline | 11 (17.5) | 0 | | | 0 | 0 |
| SAMA | Ipratropium bromide | 52 (82.5) | 0 | 19 (18.3) | 4 (3.8) | 0 | 0 |
| Xanthine | Doxofylline | 37 (58.7) | 0 | | | 0 | 0 |
| | Aminophylline | 1 (1.6) | 0 | | | 9 (17.0) | 0 |
| | Theophylline | 11 (17.5) | 0 | | | 0 | 0 |
| | Theophylline + Etofylline | 41 (65.1) | 0 | | | 0 | 0 |

We utilized the Management Sciences for Health (MSH) median international reference price for the year preceding the survey at each site. International reference prices were selected

as the most useful reference standard because they are widely available, updated frequently, and fairly stable over time. They represent actual procurement prices for medicines offered to LMICs by non-profit suppliers and international tender prices. We estimated the cost of asthma treatment based of Global Initiative for Chronic Obstructive Lung Disease (GOLD) and Global Initiative for Asthma (GINA) treatment guidelines for severity of COPD and asthma diagnosis respectively [15, 16].

## Data analysis

For comparison of pricing, we used the WHO List of Essential Medicines (WHO-EMP) and the most frequent dosing of medications available in each country setting. To facilitate international comparisons, price results are presented as median price ratios (MPR), or the ratio of a medicine's median price across outlets to the MSH [17]. We determined affordability by using standardized metrics including the total cost of a medication for a standard course of one-month's treatment (the assumed average maintenance dose per day for a drug used for its main indication in adults) based of respiratory disease severity for the lowest-wage government worker [18]. (S1 Data). For descriptive analysis frequencies and proportions were obtained for categorical variables, while means and standard deviations (SDs) were calculated for continuous variables. Paired $t$ tests were used to assess differences between groups All associations were considered statistically significance at $P$-values of $\leq 0.05$. We used ArcGIS Pro Version 2.6.2 (ESRI, Redlands, CA, USA) to develop maps of pharmacy locations within catchment areas. All data analysis was conducted in SPSS v.28 (IBM Corp, Armonk, NY, USA).

## Results

We surveyed 63 pharmacies in Nepal (95.2% private), 104 pharmacies in Peru (94.2% private) and 53 pharmacies in Uganda (98.1% private). All pharmacies provided medication data in one of the two study visits in respective catchment areas. In Fig 1, we show a map of all pharmacies within the respective catchment areas.

Public pharmacies were more likely than private pharmacies to have at least one of the respiratory medications in the survey (100%, vs 93.3%, respectively, although it should be noted that there were only six public pharmacies in Peru, three in Nepal, and one one public pharmacy in Uganda within the respective catchment areas). (Table 1) In all three sites, medications at the public pharmacies were free with a medical prescription. In Peru, to obtain the medication for free at public pharmacies, one must have the insurance type (e.g. SIS, EsSalud) that corresponds to the pharmacy where they were seeking the medication. All respiratory medications were generic in Nepal (100%) and the majority were generic in Uganda (97.2%); however among the available medications surveyed in Peru, 34.9% were brand name.

### Availability of short-acting beta agonists/short-acting muscarinic antagonists

Salbutamol (WHO-EML) monotherapy in any formulation was the most commonly available respiratory medication among the three sites (93.7% Nepal, 86.5% Peru and 79.2% Uganda). Overall, salbutamol monotherapy inhalers were available at 65.5% pharmacies across the three study sites (87.3% in Nepal, 69.2% in Peru, and 32.1% in Uganda). In Nepal, there was no significant difference in the availability of salbutamol monotherapy inhalers at private pharmacies (88.3%) compared to public pharmacies (66.6%) (p = 0.22). However, ipratropium bromide inhalers were less frequently available compared to salbutamol at 34.1% pharmacies across the three study sites (82.5% in Nepal and 22.1% in Peru) (p < 0.05). In Nepal, there were no differences in the availability of ipratropium bromide inhalers between public pharmacies (100%)

compared to private pharmacies (81.7%) (p = 0.45). Ipratropium bromide inhalers were not available at any of the pharmacies surveyed in Uganda.

## Availability of oral corticosteroids

Across all three sites, the majority of pharmacies carried oral corticosteroids (OCS), (71.4% Nepal, 87.5% Peru, 100% Uganda) and the majority of OCS was found in generic form. The majority (98.1%) of pharmacies in Uganda carried dexamethasone, and the majority (88.7%) carried prednisone. Similarly in Peru, dexamethasone and prednisone were widely available (78.9% and 86.5%, respectively). Half (50.0%) of public pharmacies in Peru had OCS where as 89.8% of private pharmacies in Peru had OCS (p = 0.05). All public pharmacies in Nepal had OCS, whereas 70.0% of private pharmacies in Nepal had OCS (p = 0.283). All public and private pharmacies surveyed in Uganda carried OCS.

## Availability of inhaled corticosteroids

Beclomethasone (WHO-EML) and fluticasone were largely unavailable in Peru (33.7%, and 5.8%, respectively) and Nepal (22.2% and, 12.7% respectively). Budesonide (WHO-EML) monotherapy was the most widely available ICS in Nepal, with 41.3% of pharmacies stocking the inhaler. Data regarding budesonide were unavailable for Peru, and no ICS were available in Uganda. In Peru, ICS were more common in public (83.3%) compared to private (30.6%) pharmacies (p = 0.05). Additionally, Nepal had the greatest formulations of combination therapy (ICS+LABA, ICS+LAMA, ICS+SABA) (Table 1).

## Availability of long-acting beta agonists

LABA monotherapy (salmeterol) was only available at 30.3% pharmacies in Nepal. In Nepal, the majority (66.6%) of public pharmacies had salmeterol with 28.3% of private pharmacies stocking salmeterol inhalers (p = 0.17). A number of ICS + LABA combination therapies were available in Nepal (87.3%, 55 total pharmacies with at least one combination) compared to Peru (6.7%, 7 total pharmacies with at least one). No pharmacies surveyed in Uganda carried any LABA therapy. In Nepal, 88.3% and 66.6% private and public pharmacies, respectviely, stocked at least one ICS + LABA combination therapy (p = 0.22). In Peru, 7.1% private pharmacies had ICS + LABA inhalers available, with no public pharmacies stocking the medication (p = 0.50).

## Availability of long-acting muscarinic agonists

Tiotropium monotherapy (WHO-EML) was only available in Nepal (77.8%). In Nepal, 78.3% private and 66.6% of public pharmacies stocked tiotropium monotherapy (p = 0.60). Formoterol + tiotropium combination therapy (LABA + LAMA, WHO-EML) was available among 14.3% of pharmacies in Nepal. 13.3% of private and 100% of public pharmacies in Nepal reported having LABA+LAMA inhalers stocked (p = 0.05). LAMA was available only in generic formulations in Nepal.

## Availability of other respiratory medications

Xanthine-based respiratory medications were more common in Nepal (77.8%) compared to Uganda (17.0%), with the majority of pharmacies (50.0%) carrying at least one formulation (Aminophylline, Doxofylline, Theophylline, Theophylline + Etofylline). Among the two sites where antibiotics were surveyed (Nepal and Uganda), the majority of pharmacies carried at

least one antibiotic as part of standard management for a respiratory exacerbation. In Uganda and Nepal, the majority of antibiotics were generic (100% and 99.8% respectively).

## Affordability of inhaled medications

Affordability of medications across sites in terms of days' wages was variable (Table 2). Nepal had the lowest monthly retail cost of all inhaled medications as well as the lowest impact in terms of days' wages needed per treatment. The cost of treatment for mild intermittent asthma in private pharmacies ranged from 0.47 in Peru and Nepal to 3.3 days' wages Uganda, although these calculations were based on limited data due to low availability of inhaler medications. Treatment for severe COPD, defined as a diagnosis of COPD with significant functional limitations and frequent exacerbations, accounted for 2.38 days wages in Nepal. Nepal similarly had the lowest MPR across a range of medication catagories. (Table 3) The cost of salbutemol inhalers was slightly above the median international reference unit price (0.010), although the MPR of combination inhalers (ICS-LABA, LABA-LAMA) varied significantly (5.63 in Peru to 13.503 in Nepal). Uganda had the highest MPR across medication classes.

## Discussion

We conducted a survey of respiratory medication availability and affordability among pharmacies in three distinct geographic LMIC settings using the WHO/HAI methodology [11]. The availability of respiratory medications varied across the three settings (Fig 2). While salbutamol

**Table 2. Affordability of selected medicines to treat asthma and COPD based on mean total monthly cost and number of days' wages needed for a minimum wage government worker.**

| Condition | Treatment Schedule | Nepal | | Peru | | Uganda | | P-value |
|---|---|---|---|---|---|---|---|---|
| | | Mean Total Monthly Cost (USD) | Days' wages to pay treatment | Mean Total Monthly Cost (USD) | Days' wages to pay treatment | Mean Total Monthly Cost (USD) | Days' wages to pay treatment | |
| **Asthma** | | | | | | | | |
| **Intermittent** 100 mcg Salbutamol/ Albuterol inhaler | 1 inhaler per month | $1.90 | 0.47 days | $3.94 | 0.47 days | $3.08 | 3.33 days | 0.012 |
| **Moderate Persistent** 25 + 250 mcg Fluticasone + Salmeterol inhaler | 1 inhaler per month | $7.09 | 1.75 days | $17.88 | 2.12 days | --- | --- | 0.0058 |
| **Severe Persistent** 50 + 250 mcg Fluticasone + Salmeterol inhaler | 1 inhaler per month | $4.52 | 1.12 days | --- | --- | --- | --- | |
| **COPD** | | | | | | | | |
| **GOLD A** 100 mcg Salbutamol/ Albuterol inhaler | 1 inhaler per month | $1.90 | 0.47 days | $3.94 | 0.47 days | $3.08 | 3.33 days | 0.012 |
| **GOLD B** 9 mcg Tiotropium inhaler | 1 inhaler per month | $7.05 | 1.74 days | --- | ---- | --- | --- | |
| **GOLD C** 9 mcg Tiotropium inhaler | 1 inhaler per month | $7.05 | 1.74 days | --- | --- | --- | --- | |
| **GOLD D** 9 mcg Tiotropium inhaler 25 mcg Salmeterol inhaler | 2 inhalers per month | $9.63 | 2.38 days | --- | --- | --- | --- | |

Affordability calculated by using standardized metrics including the total cost of a medication for a standard course of one-month's treatment to the daily wage of the lowest paid unskilled government sector worker based of respiratory disease severity.

**Table 3. Medication Median Price Ratio (MPR).**

| Medication Category and Medication | | Median Price Ratio | | |
|---|---|---|---|---|
| | | **Nepal** | **Peru** | **Uganda** |
| **Antibiotics** | 250 mg Amoxicillinpill | | | 1.248 |
| | 500+125 mg Amoxicillin + Clavulanic Acidpill | | | 3.978 |
| | 250 mg Azithromycin pill | | | 0.954 |
| | 250 mg Ciprofloxacin pill | | | 2.179 |
| | 100 mg Doxycyclinepill | 0.260 | | 2.951 |
| | 500 mg Levofloxacinpill | 0.102 | | 2.989 |
| **Inhaled Corticosteroids (ICS)** | 250 mcg Beclomethasone inhaler | | 1.802 | |
| | 200 mcg Budesonide inhaler | 0.671 | | |
| **ICS + LABA** | 25+250 mcg Fluticasone + Salmeterolinhaler | 13.503 | 5.629 | |
| **Oral Corticosteroids** | 0.5 mg Dexamethasone | 0.004 | 58.373 | 2.156 |
| | 5 mg Prednisone | | 0.237 | 2.249 |
| **SABA** | 100 mcg Salbutamol/Albuterol inhaler | 0.010 | 2.315 | 2.442 |
| **Xanthine** | 100 mg Aminophyllinepill | | | 0.457 |
| | 200 mg Theophyllinepill | 0.398 | | |

MPR calculated as the ratio of a medicine's median price across outlets to the Management Sciences for Health median international reference price for the year preceding the survey at each site

was the only respiratory medication available across all study settings, with the lowest MPR, it was unaffordable based on daily wages in rural Uganda. ICS were available in pharmacies in Peru and Nepal and carried in all public pharmacies surveyed. Nepal overall had the highest availability and variety of respiratory medications, which on average had the lowest MPR across catagories. Peru and Uganda had similar MPR for salbutamol, though the affordability varied based on the number of days' wages needed for a minimum wage worker in each site.

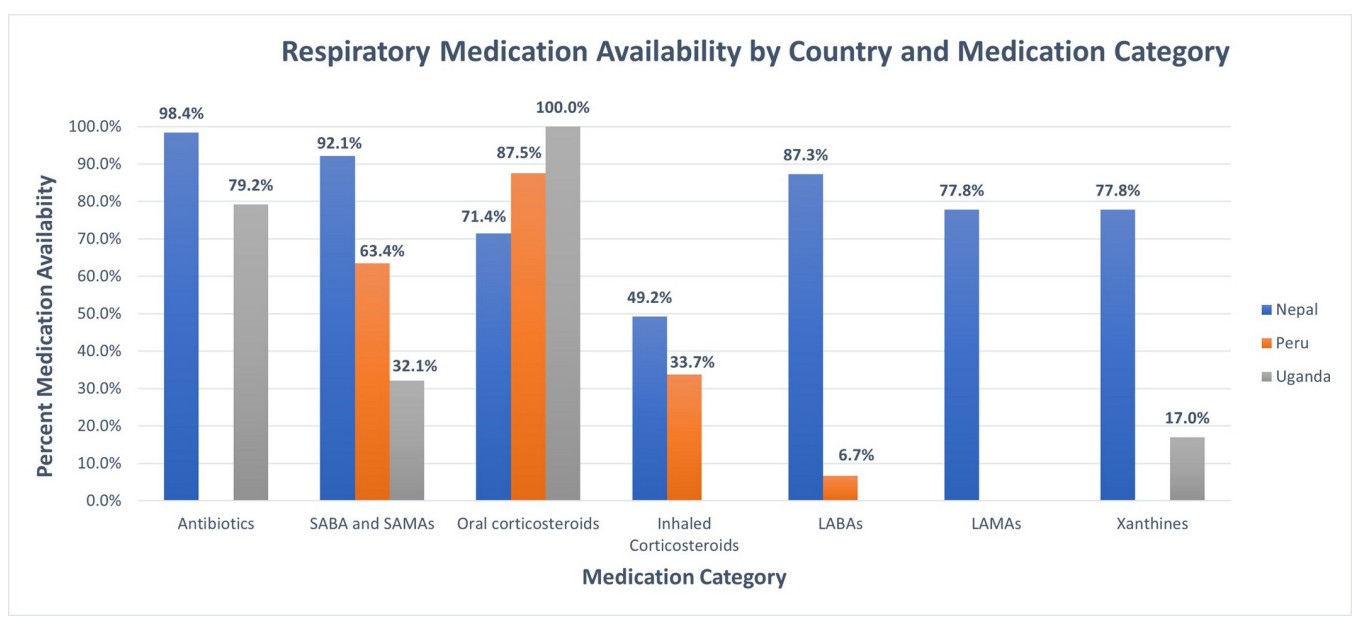

**Fig 2. Percent availability of respiratory medicine stratified by medication category.** Percent availability compared across sites by medication category. Note LAMAs, Antibiotics and Xanthines not surveyed in Peru.

We found short-acting medications used for symptomatic management were more widely available and affordable compared to long-acting bronchodilator and inhaled corticosteroid medications used for chronic management. In Uganda, COPD long-term treatment were not available at surveyed pharmacies in our study, indicating a lack of access to essentials medications to manage COPD, where the prevalence has been reported as high as 16.8% in rural Ugandan residents [19, 20]. Daily maintenance therapy for asthma and COPD has previously been shown to decrease exacerbations, improve quality of life, and improve lung function decline with significant economic impact [21, 22]. The study sites in Nepal and Peru, middle-income countries, had increased availability and affordability of respiratory medications compared the site in Uganda, a low-income country. This further points to a disparity in respiratory medication access between middle- and low-income countries. Prioritizing long-term management may decrease the financial burden and improve healthcare outcomes for asthma and COPD in LMICs.

Surveys of medication availability vary in their scope, with some including a few pharmacies over large geographic areas, and others focused on in-depth assessments of smaller geographic areas. The World Health Organization (WHO)/Health Action International (HAI) developed standardized survey instruments and sampled pharmacies within and around the countries' major urban centers, allowing for price comparisons [13]. Kibirige and colleagues conducted an availability and cost survey of 130 pharmacies across Uganda; however these were tertiary hospitals in close proximity to each other [23]. Additional surveys and reviews have demonstrated high degree of variability among the sites studied with similar lack of long acting and combination inhalers [11, 24, 25].

Our findings are similar with previous studies demonstrating high cost and low affordability in Sub-Saharan Africa compared to South America and Southeast Asia [6]. Variation in MPR can be related to several factors. In Nepal, medicines are exempt from value-added tax and central government tax [9]. While public sector clinics provide access to free medications, availability has historically been limited and the majority of health care financing consists of out of pocket costs [9, 26]. Nepal has the highest variety of types of generic respiratory medicines, additionally allowing for pricing competition and lower overall prices [27]. This high variation is likely due to Nepal's proximity to India, where 95% of Nepal's medication supply is manufactured [28]. The availability of medications beyond the EML, as well as combination therapy has the potential to result in inappropriate or inadequate treatment for respiratory diseases. In many settings, combination antibiotics with respiratory medications can result in resistance patterns [29].

There are several documented barriers and challenges that lead to variation in the costs of essential medications in LMICs. Most respiratory medications are administered by inhalers or are nebulized. These technologies present challenges as they are among the more complex devices manufactured by the pharmaceutical industry, preventing addition barriers to the production of generics, and requiring reliable delivery of active medications to the lower airways [6]. Institutional, administrative, and legal barriers lead to onerous registration of medications and increased transaction costs and pricing. Limited procurement capacity combined with fragmentation of demand may similarly lead to smaller purchase quantities, thereby discouraging suppliers from entering low-volume markets. In settings where most health expenditures are out of pocket, efficient procurement is essential for availability and affordability in both public and private sector markets. Standardization of treatment guidelines and purchasing practices can streamline medication procurement, leading to lower costs [6, 30]. Additionally, in many LMIC settings, generic or publicly provided medications can be viewed as inferior to brand name versions [31, 32]. The availability of guidelines varies between LMIC settings, and GOLD does not currently provide specific guidance for resource-constrained settings [4, 33].

Even where guidelines are available, there are recognized barriers to effective implementation of guideline-based care, which include but are not limited to access to affordable medications.

There are several policy implications of these findings. First, improved access to primary care allows for initiation of treatment for a range of chronic respiratory diseases and demand for medications. Pooled procurement of WHO EML medications can substantially reduce costs. Decentralized distribution systems in rural settings have been implemented with success for communicable diseases like HIV and tuberculosis and may be a promising model for delivery of medications for non-communicable diseases, as well [34]. Industry best practices can be compiled in the areas of packaging, pricing, and patient education to achieve better drug treatment adherence. Improvements in supply chain integrity and pooled quality assessment at port of entry can reduce additional mark up and decrease counterfeited medications.

The study aimed to conduct surveys of all pharmacies within the allocated areas. A strength of this study is the inclusion of rural and urban sites, as well as diverse socioeconomic characteristics of the regions allowing for sampling of distinct geographic areas and direct assessment of COPD prevalence in those areas. Prior studies limited sampling to a few pharmacies in mainly urban areas, potentially biasing findings. The present study, however, was localized to specific areas within Peru, Nepal and Uganda, and therefore are not representative of the entire country. We conducted surveys of standard and essential respiratory medications, as well as antibiotics used for the treatment of respiratory infections, allowing for a complete list of medication classes in each site. A limitation of the study was that certain medications (e.g. LAMAs, Xanthine-based medications) were not surveyed in Peru, thus limiting comparisons with the other sites. Additionally, leukotriene antagonists were not surveyed. Another important limitation of this study is that the data in Peru were collected two years prior to those in Uganda and Nepal. It is possible that the availability of these medications could have changed during this two-year period. There were no changes in the Peruvian National Essential Medicines List among the medications surveyed in Peru.

The WHO set a target of 80% availability of essential affordable medications by 2025. However, there is high variability in access and affordabilty of respiratory medicines across geographic settings. Policies aimed at decreasing value-added taxes, promoting generic medications, and standardizing procurement may lead to improved availability and affordability in LMIC settings.

## Supporting information

**S1 Data. Site specific information.**
(DOCX)

## Acknowledgments

**GECo study investigators:** *University College London*, Shumonta Quaderi MBBS BSc, Susan Michie Dphil Phil Cpsychol, John R Hurst PhD FRCP, Zachos Anastasiou, Julie Barber PhD

*University of Miami*, Trishul Siddharthan MD *Johns Hopkins University*, Suzanne L Pollard PhD, MSPH, Natalie A. Rykiel MSc, Nicole Robertson, William Checkley MD PhD *Institute of Medicine*, Laxman Shrestha MD, Karbir Nath Yogi MD, Arun Sharma MD *Siddhi Memorial Hospital*, Ram K Chandyo *Makerere University*, Patricia Alupo MB ChB Mmed, Denis Muwonge, Denis Mawanda, Faith Nassali, Robert Kalyesubula MB ChB Mmed, Bruce Kirenga MB ChB Mmed *University of York*, Andrew J Mirelman PhD, MPH, Marta Soares MSc *A.B. PRISMA*, Oscar Flores-Flores MD, Elisa Romani-Huacani *CRONICAS Centre of Excellence in Chronic Diseases*, *Universidad Peruana Cayetano Heredia*, Maria Kathia Cárdenas MSc, J.

Jaime Miranda, MD PhD *University of California San Francisco*, Adithya Cattamanchi MD, MAS

## Author Contributions

**Conceptualization:** Trishul Siddharthan, Nicole M. Robertson, Natalie A. Rykiel, Bruce Kirenga, Jing Luo, J. Jaime Miranda, William Checkley, John R. Hurst, Suzanne L. Pollard.

**Data curation:** Trishul Siddharthan, Nicole M. Robertson, Lindsay J. Underhill, Roma Padalkar, Sarah McKeown, Suzanne L. Pollard.

**Formal analysis:** Trishul Siddharthan, Nicole M. Robertson, Natalie A. Rykiel, Lindsay J. Underhill, Nihaal Rahman, Sakshi Mohan, Roma Padalkar, Oscar Flores-Flores, Suzanne L. Pollard.

**Funding acquisition:** Trishul Siddharthan, Bruce Kirenga, J. Jaime Miranda, William Checkley, John R. Hurst, Suzanne L. Pollard.

**Investigation:** Trishul Siddharthan, Oscar Flores-Flores, Shumonta A. Quaderi, Maria Kathia Cárdenas, Suzanne L. Pollard.

**Methodology:** Trishul Siddharthan, Suzanne L. Pollard.

**Project administration:** Trishul Siddharthan, Sujan Kafle.

**Supervision:** Sarah McKeown, Oscar Flores-Flores, Shumonta A. Quaderi, Bruce Kirenga, J. Jaime Miranda, John R. Hurst.

**Writing – original draft:** Trishul Siddharthan, Natalie A. Rykiel, Roma Padalkar, Sarah McKeown, William Checkley, John R. Hurst, Suzanne L. Pollard.

**Writing – review & editing:** Trishul Siddharthan, Nicole M. Robertson, Natalie A. Rykiel, Lindsay J. Underhill, Nihaal Rahman, Sujan Kafle, Sakshi Mohan, Roma Padalkar, Sarah McKeown, Oscar Flores-Flores, Shumonta A. Quaderi, Patricia Alupo, Robert Kalyesubula, Bruce Kirenga, Jing Luo, Maria Kathia Cárdenas, Gonzalo Gianella, J. Jaime Miranda, William Checkley, John R. Hurst, Suzanne L. Pollard.

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
