## [Decision Letter · Decision Letter 0]

26 Jul 2022

PGPH-D-22-00869

Availability, Affordability and Access to Essential Medications for Asthma and Chronic Obstructive Pulmonary Disease in Three Low- and Middle-Income Country Settings

Dear Dr. Siddharthan,

Thank you for submitting your manuscript to PLOS Global Public Health. After careful consideration, we feel that it has merit but does not fully meet PLOS Global Public Health’s publication criteria as it currently stands. Therefore, we invite you to submit a revised version of the manuscript that addresses the points raised during the review process.

This manuscript addresses an important public health issueon"Availability, Affordability and Access to Essential Medications for Asthma and Chronic Obstructive Pulmonary Disease in Three Low- and Middle-Income Country Settings". However, this requires "Major Revisions" undermethodology and discussion sections of the manuscript.

We look forward to receiving your revised manuscript.

Kind regards,

Laila Akbar Ladak, PhD, MScN, BScN, RN

Section Editor

Journal Requirements:

Additional Editor Comments (if provided):

Reviewers' comments:

Reviewer's Responses to Questions

**Comments to the Author**

1. Does this manuscript meet PLOS Global Public Health’s publication criteria? Is the manuscript technically sound, and do the data support the conclusions? The manuscript must describe methodologically and ethically rigorous research with conclusions that are appropriately drawn based on the data presented.

Reviewer #1:Yes

Reviewer #2:Yes

Reviewer #3:Partly

2. Has the statistical analysis been performed appropriately and rigorously?

Reviewer #1:Yes

Reviewer #2:Yes

Reviewer #3:No

3. Have the authors made all data underlying the findings in their manuscript fully available (please refer to the Data Availability Statement at the start of the manuscript PDF file)?

Reviewer #1:Yes

Reviewer #2:Yes

Reviewer #3:Yes

4. Is the manuscript presented in an intelligible fashion and written in standard English?

Reviewer #1:Yes

Reviewer #2:Yes

Reviewer #3:Yes

5. Review Comments to the Author

Reviewer #1:Reviewer’s comments

Manuscript title: Availability, Affordability and Access to Essential Medications for Asthma and Chronic Obstructive Pulmonary Disease in Three Low- and Middle-Income Country Settings.

This study by Siddharthan T et al is a well-written study highlighting an important area of inequity in access to essential medicines for asthma and COPD in three low-and middle-income countries. It comprehensively assesses the availability, pricing, and affordability of essential medicines as recommended by the WHO essential medicines list.

Despite its strengths, there are areas that need to be revised in the manuscript to improve its quality.

Major revisions.

1. Information on the availability of LAMA and LABA combination, a key medicine in the management of COPD, is missing in the abstract. I would advise that this information is added.

2. There is little reference to studies evaluating the availability and affordability of asthma and COPD medicines performed in similar LMIC settings in the discussion section. The authors should include some of these key studies below to improve their discussion section:

- Kibirige D et al. Availability and affordability of medicines and diagnostic tests recommended for management of asthma and chronic obstructive pulmonary disease in sub-Saharan Africa: a systematic review. Allergy Asthma Clin Immunol. 2019 Mar 7;15:14.

- Mendis S, Fukino K, Cameron A, Laing R, Filipe A Jr, Khatib O, Leowski J, Ewen M. The availability and affordability of selected essential medicines for chronic diseases in six low- and middle-income countries. Bull World Health Organ. 2007;85:279–88.

- Cameron A, Ewen M, Ross-Degnan D, Ball D, Laing R. Medicine prices, availability, and affordability in 36 developing and middle-income countries: a secondary analysis. Lancet. 2009;373:240–9.

- Mendis S, Al Bashir I, Dissanayake L, Varghese C, Fadhil I, Marhe E, Sambo B, Mehta F, Elsayad H, Sow I, et al. Gaps in capacity in primary care in low-resource settings for implementation of essential noncommunicable disease interventions. Int J Hypertens. 2012;2012:584041.

- Ozoh OB et al. Asthma and COPD medicines availability and affordability in Nigeria. Trop Med Int Health. 2021 Jan;26(1):54-65.

3. The authors should include information about the wages for the lowest paid unskilled government worker that was used to assess the affordability of these medicines in each country/study setting.

4. In relation to my above comment, I find the reported affordability of salbutamol inhaler as treatment of intermittent asthma very inaccurate.

The authors should refer to the Ministry of Public Service, Republic of Uganda website and this link below for guidance on the wage structure in 2019/2020, a period when the study was conducted: https://www.publicservice.go.ug/media/resources/Salary%20Structure%20FY%20201819%20Schedule%201%20-%2012.pdf

According to the Public Service wage structure in Uganda, the lowest-paid unskilled government worker is at the U8 level. Using that information, please revise your results on the affordability of the drug in Uganda.

5. There is a study limitation of lack of information on the availability and affordability of leukotriene antagonists which are key medicines in the management of asthma.

Minor comments

The manuscript has several typographical errors that need extensive revision.

An example of these errors is shown below:

- On the 2nd last line of page 8, the number of health facilities where Salmeterol was present is missing (30.3%, n=……….).

- On page 9, Formoterol is misspelled (section on availability of long-acting antimuscarinic agents).

- On page 10, the author’s name is misspelt (Kibirige NOT Kabirige).

- Affordability should be corrected and expressed as days’ wages throughout the manuscript (see the section on the affordability of essential medicines and table 2).

Reviewer #2:General

While I agree that such availability studies are very useful for a given country, I am less convinced that such a cross-country study has much additional benefit. Also, for this study, the question comes up: so what? It seems that the only conclusion is that the availability is usually higher in private pharmacies (with a few exceptions) and that prices vary between the three countries; and that the range of medicines in Nepal seems wider. So what can we do with this information?

I would recommend that you try to expand the discussion a little further, and formulate hypotheses to explain the large price differences between the three countries. It could also be discussed why there are so many combination medicines in Nepal – I am rather sure that very few of such combinations are on the WHO Model List. It would also be very interesting if you formulated some practical advice to each of the three countries about how to improve their particular situation.

Methods

What sort of region were these – urban, semi-urban, rural? Please give some additional information

Sample seslection: I am not very sure about how the medicines included in the survey were selected – especially in view of the many non-essential combinations used in Nepal. Did you measure and report any drug in the selected categories that you could find in the private pharmacies surveyed? Why did you not limit yourself to medicines on the WHO Model List? I would not like the paper to suggest that the many non-essential medicines observed as available in Nepal should actually have been available in the other two countries. At least, in the discussion I recommend that you reflect on the differences between the WHO Model List and the medicines actually found in the pharmacies in the three countries.

Data analysis

Line 12: Check grammar

Iine 14: GOLD and GINA first time in full

Results: There is no need to describe all figures and percentages again as text as they read better as tables. So I suggest to remove all the “drug – specific” text subsections and restrict yourself to referring to the two tables, and (perhaps) mentioning one or two specific important facts.

Results: Think the number of public pharmacies studied is so small (both absolutely as relatively) that I doubt that you can make a meaningful comparison between the public and private sector. Can you please check your statistics whether any differences observed are statistically signifiant?

Reviewer #3:This study is to provide information on availability, affordability, and access to essential medicines for selected respiratory diseases in three countries. Below are a few major concerns:

1. Introduction: Although the manuscript highlighted the scale of respiratory disease, there is little information on (1) reasons why it is important to study availability, affordability, and access in the three countries. In another word, what is the uniqueness of this study; (2) what has been done previously in the three countries; and (3) what gaps would this study fill in?

2. Methods: The methodology to carry out the study and analysis is not adequately described. The method sector also needs to be restructured, including study design, sampling, measurement, data collection and analysis.

2.1. The selection of countries and catchment areas is not described in detail, particularly the selection of the catchment areas. how were they selected? Were they selected through random sampling? were they representative of the country's catchment areas?

2.2. There is no description of the drug system in the three countries to prepare readers to understand the study setting, such as the pharmaceutical market share between the public and private sectors.

2.3. The measurements should be described in more detail under a separate heading. The manuscript should be clear about how availability, affordability, and access were measured. The measure should not be under the data analysis. The development of indicators should also be described in more detail. Key parameters should be provided, such as the daily wage for each country.

2.4. Most statistical analysis is descriptive. Further analysis could be explored, such as the comparison of availability and affordability among sites and drugs, and between sectors.

3. Results

3.1. suggest conducting further analysis to provide comparative analysis.

3.2. Table 2. The days to pay for treatment for Uganda are extremely high. What is the daily wage used for comparisons? According to https://wageindicator.org/salary/minimum-wage/uganda, the minimum wage per month is UGX 130,000, which is equivalent to $34.38 per month or 1.15 per day. The mean total days for Uganda would be about 3 days, rather than 54 days.

4. discussion

4.1. The discussion should focus more on explaining why there is a shortage in some countries for particular medicines and how it is related to the drug delivery systems.

6. PLOS authors have the option to publish the peer review history of their article (what does this mean?). If published, this will include your full peer review and any attached files.

**Do you want your identity to be public for this peer review?** For information about this choice, including consent withdrawal, please see our Privacy Policy.

Reviewer #1:No

Reviewer #2:**Yes:**Prof Dr Hans V Hogerzeil

Reviewer #3:No

---

## [Decision Letter · Decision Letter 1]

31 Oct 2022

Availability, Affordability and Access to Essential Medications for Asthma and Chronic Obstructive Pulmonary Disease in Three Low- and Middle-Income Country Settings

PGPH-D-22-00869R1

Dear Dr. Siddharthan,

We are pleased to inform you that your manuscript 'Availability, Affordability and Access to Essential Medications for Asthma and Chronic Obstructive Pulmonary Disease in Three Low- and Middle-Income Country Settings' has been provisionally accepted for publication in PLOS Global Public Health.

Best regards,

Hassan Haghparast Bidgoli

Academic Editor

Thank you for addressing the reviewers' comments. Except for the minor edit by reviewer 2 (changing the word "counterfeit" into "falsified"), there is no further comments from the reviewers and editor.

Reviewer Comments (if any, and for reference):

Reviewer's Responses to Questions

**Comments to the Author**

1. If the authors have adequately addressed your comments raised in a previous round of review and you feel that this manuscript is now acceptable for publication, you may indicate that here to bypass the “Comments to the Author” section, enter your conflict of interest statement in the “Confidential to Editor” section, and submit your "Accept" recommendation.

Reviewer #1:All comments have been addressed

Reviewer #2:All comments have been addressed

2. Does this manuscript meet PLOS Global Public Health’s publication criteria? Is the manuscript technically sound, and do the data support the conclusions? The manuscript must describe methodologically and ethically rigorous research with conclusions that are appropriately drawn based on the data presented.

Reviewer #1:Yes

Reviewer #2:Yes

3. Has the statistical analysis been performed appropriately and rigorously?

Reviewer #1:Yes

Reviewer #2:I don't know

4. Have the authors made all data underlying the findings in their manuscript fully available (please refer to the Data Availability Statement at the start of the manuscript PDF file)?

Reviewer #1:Yes

Reviewer #2:Yes

5. Is the manuscript presented in an intelligible fashion and written in standard English?

Reviewer #1:Yes

Reviewer #2:Yes

6. Review Comments to the Author

Reviewer #1:All comments that I raised have been adequately addressed.

Reviewer #2:Thank you for considering my reviewer's comments. My only advise is to change the word "counterfeit" into "falsified" which is the correct term used by WHO (counterfeit is only a trademark issue, falsified is a real falsification as meant here)

7. PLOS authors have the option to publish the peer review history of their article (what does this mean?). If published, this will include your full peer review and any attached files.

**Do you want your identity to be public for this peer review?** For information about this choice, including consent withdrawal, please see our Privacy Policy.

Reviewer #1:No

Reviewer #2:**Yes:**Hans V Hogerzeil
